# Physical Characterization of Ten Hemp Varieties to Use as Animal Bedding Material

**DOI:** 10.3390/ani13020284

**Published:** 2023-01-13

**Authors:** Sheyla Arango, Rosa Greco, Nadia Guzzo, Emiliano Raffrenato, Massimo Montanari, Lucia Bailoni

**Affiliations:** 1Department of Comparative Biomedicine and Food Science (BCA), University of Padova, Viale dell’Universitá 16, 35020 Legnaro, Italy; 2Department of Land, Environment, Agriculture and Forestry (TESAF), University of Padova, Viale dell’Universitá 16, 35020 Legnaro, Italy; 3Council for Agricultural Research and Economics, Research Centre for Cereal and Industrial Crops, Via di Corticella 133, 40128 Bologna, Italy

**Keywords:** hemp, hemp shives, bedding material

## Abstract

**Simple Summary:**

Hemp hurd is the inner bark of the plant’s stem and is typically seen as waste. The objective of this experiment was to describe the physical characteristics, such as moisture, water absorption, and ammonia absorption of ten hemp varieties (Fibranova, Codimono, USO31, CS, Futura 75, Eletta Campana, Carmaleonte, Felina 32, Santhica, and Ferimon) cultivated in Italy to be used for livestock bedding. The samples were hemp shives, obtained from hemp hurds ground to 8 mm. The results showed an average of 7.78%, 251.9%, and 50.0% for moisture, water absorption, and ammonia absorption, respectively. Moisture and water absorption were similar among varieties. A significant difference between varieties was found for the ammonia absorption, varying from 45.0 to 55.5% for the Fibranova and Ferimon, respectively. In conclusion, hemp shives have good physical characteristics, similar to other commercial bedding materials (i.e., wood shavings) but other parameters and on-farm trials will be required to make a full assessment of hemp.

**Abstract:**

Hemp (*Cannabis sativa* L.) hurds, the inner bark of the stem, are a poorly appreciated part of the plant that typically represents waste. The aim of this experiment was to describe the physical characteristics, including moisture (M), water absorption (WA), and ammonia absorption (AA), of 10 hemp varieties (Fibranova, Codimono, USO31, CS, Futura 75, Eletta Campana, Carmaleonte, Felina 32, Santhica, and Ferimon) cultivated in Italy. Samples of hemp hurds were ground to 8 mm obtaining hemp shives. Values of M, WA, and AA were determined following the official procedures. The results showed an average of 7.78%, 251.9%, and 50.0% for M, WA, and AA, respectively. Data of M and WA were similar among varieties, whereas a significant difference was found for the AA, varying from 45.0 to 55.5% for the Fibranova and Ferimon varieties, respectively. In conclusion, hemp shives have good physical characteristics, similar to other commercial bedding materials (i.e., wood shavings) but other parameters and on-farm trials will be required to make a full assessment of hemp.

## 1. Introduction

Several materials have been used for livestock bedding. A good bedding material should provide animal comfort, have good absorption capacity of water and ammonia, decompose quickly with manure, be economic and not cause hygiene problems [1,2,3,4,5]. Among the popular bedding materials are crop straws, wood shavings, peat, seed hulls, and corn stover [2,6,7,8,9,10]. Crop straws are the fibrous residue from grain crop harvest and may be one of the most common bedding materials used in farms of Central Europe and southern Nordics countries [1,3]. Wood shavings are used worldwide, mostly because it is one of the cheapest options in the market [1,4]. Peat is a quite common bedding material, especially in countries like Finland and Sweden [1]. Seed hulls, mostly represented by rice hulls is an important by-product of the rice milling process. Corn stover, which is a by-product of the processing of the most important cereal of the world [3]. Even though we have all these well know bedding materials, some other crop residues are always being tested as a simple way to valorize agro-waste resources [11] because they are continuously generated in large quantities all over the world [12].

Hemp (*Cannabis sativa* L.), an ancient plant primarily cultivated for its fiber, has great added value because each part of the plant represents many potentially valuable resources for quality products [13]. As a muti-purpose crop, hemp delivers: fibre, hurds, seeds, flowers and leaves. Fibre is used for the obtention of paper, biocomposites and insulation materials. Hurds are used for construction and animal bedding. Seeds have a high nutritional value with an excellent and unique fatty acid profile so it is used to produce oil and other by-products. Flowers and leaves processing lead to the obtention of pharmaceutical and food supplements that contain non-psychotropic cannabinoid (CBD) and it is used for medical purposes [14]. The diversity of hemp is shown by the 70 varieties included in the EU Common Catalogue of Varieties of Agricultural Plant Species that can be divided into two broad categories, those suited for seed production and those for fiber production. Among the seed varieties, Felina 32 and Ferimon are included. Whereas, CS, Fibranova, Eletta Campana, Futura 75 and Santhica are considered fiber varieties. They are generally taller (1–5 m) [12], and exhibit optimum fiber yields when cultivated in temperate climates with an annual rainfall on average of 630–750 mm [15]. In Italy, hemp is sown in spring (April–May) and harvested in autumn (September–October). The harvesting of plants for fiber production (roughly 70–90 days after sowing) is preferably made at the flowering stage, as further maturation increases the proportion of undesirable “secondary” bast fibers in plants [15]. Whether hemp’s main purpose is to obtain seeds or fiber, hemp hurds always end up as a sub-product [16]. In fact, the relation between hemp hurds and fibers is 1.7 to 1 [16]. Hemp hurds are basically the inner bark of the stem, which is the hemp core or the leftover bast fiber that typically contains around 20 to 30% of lignin. They are a poorly appreciated part of the plant which typically ends up as landfill [17], embedded in the ground or, more recently, if collected, used in green buildings [12]. 

As hemp production is increasing in Europe, along with the global need to guarantee sustainable crop management using zero-waste strategies, the evaluation of using hemp hurds as a potential bedding material is needed. At present, hemp hurds are considered one of the most interesting waste products obtained from hemp [16]. As they can absorb moisture up to four times their dry weight, they have already reached 63% of market participation as an animal bedding material for horses and other farm animals like chickens [14]. However, no studies are available on the physical properties of hemp, such as moisture content, water absorption, or ammonia absorption to increase its use in this field. Therefore, this study evaluated 10 different hemp varieties cultivated in Italy in order to verify the main physical parameters and define their viability for livestock bedding. 

## 2. Materials and Methods

### 2.1. Test Materials

Ten varieties of *Cannabis sativa* L. were evaluated: Fibranova, Codimono, USO31, CS, Futura 75, Eletta Campana, Carmaleonte, Felina 32, Santhica, and Ferimon. They were cultivated at the Center for Cereal and Industrial Crops (CREA-CI), located in Rovigo (Veneto Region, Northern Italy). 

### 2.2. Hemp Sample Obtention

After harvesting, 30 plants were randomly selected from each variety. The separation of the fiber was carried out at CREA-CI (Rovigo, Italy) and the final step for the sample obtention was made at the Bio-fuel Analysis Laboratory (ABC Laboratory) of the Department of Land, Environment, Agriculture and Forestry (TESAF) of the University of Padua.

The central part of the stem was cut to obtain 60 cm stalks that were left in the sun to dry naturally. The stalks were totally immersed under water and remained at a temperature of 30–35 °C for 7 days. Then, each stalk was rinsed with clean water until the mucilage was completely removed and the fiber separated (Figure 1a). 

After this, they were dried in the sun for 24 h and then dried in an oven at 60 °C for 3 days until a constant weight was obtained. Finally, they were processed using a cutting mill (SM 100 RETSCH GmbH, Haan, Germany) equipped with an 8 mm sieve in order to obtain the hemp shives (Figure 1b).

### 2.3. Physical Measurements

Analyses were carried out in the ABC Laboratory, where moisture content (M), water absorption (WA), and ammonia absorption (AA) were analyzed. For the moisture content, three repetitions of each variety were measured following the standard procedure of UNI EN ISO 18134-1 (2015). A 300 g sample was oven dried at 105 °C. Every 60 min, the samples were taken from the oven and weighed until two consecutive weights were found to be stable (with under 0.2% variation allowed). Then, M was calculated from the difference in the sample weight before and after drying and expressed as a percentage. The WA was assessed following the procedure of Potgieter and Wilke (1996) [18], in which, 150 mL of water and 10 g of sample were left to soak for one hour in a closed filter funnel. Then, the funnel was opened to drain the excess water through filtration. The AA followed the procedure of Fleming et al. (2008) [19], but ammonia 9% was used instead of the mixture of horse urine and feces.

### 2.4. Statistical Measurements

Statistical analysis was conducted using PROC GLM procedure in SAS (SAS Institute Inc., Cary, NC, USA, 2009). Data from M, WA, and AA were analyzed with a monofactorial model that considered the effect of variety (10 levels). For all the variables, the comparisons between LS means were performed using the Tukey test, and differences were considered significant at *p* < 0.05.

## 3. Results

### 3.1. Moisture Content

All the values of moisture content were similar (*p* > 0.05) among varieties (Table 1). The mean value was 7.78 ± 0.29%.

### 3.2. Water Absorption

No statistical differences (*p* > 0.05) were observed among varieties (Table 1), even if the range between the highest value (317.9%; Felina 32) and the lowest value (211.4%; Eletta Campana) was very wide.

### 3.3. Ammonia Absorption

The values of ammonia absorption (Table 1) ranged from 45.0% (Fibranova) to 55.5% (Ferimon). The differences among varieties were statistically significant (*p* < 0.05).

## 4. Discussion

As no previous study of hemp as material for animal bedding has been carried out before, the whole method of processing the hemp to obtain the samples and the methods used to analyze the physical characteristics were difficult to choose but are essential to understand and explain our results. Hemp hurds were chosen because the hemp fiber industry obtains dust-free hemp hurds as a waste product that can be directly used for livestock bedding. Indeed, there is already a stable market for this commodity, mostly for pets and horses [20]. Starting from the decortication process, which is the separation of fibers from hurds, we can already point out some aspects to consider. The type of process applied for hemp fiber extraction has an effect on the chemical composition of the fibers and the resulting properties not only of the fibers themselves but also of the hurds obtained as a co-product. For this experiment, the method of fiber extraction was similar to that called osmotic degumming [21]. Of course, the different methods of fiber extraction made on a large scale by hemp factories will change the quality of hurds and their properties considerably [21]. After obtaining the hurds, a further step was needed to transform them into an appealing bedding material. In order to do that, we used a mill to turn the hemp hurds into hemp shives. It is clear that the particle size obtained after the whole processing of the hurds influenced all the physical parameters reported in this study. For instance, small particles usually give better performance for water absorption, because of the increased ratio of surface to volume [22]. 

Our results showed that the moisture content between hemp varieties was not significantly different. This is mostly because they come from the same field and been under the same storage conditions before the beginning of the experiment. Moisture content is an important factor to consider in the choice of any bedding material. High moisture in the bedding increases ammonia build-up through increased microbial metabolism, resulting in respiratory lesions [18], whereas a low moisture content assures a longer storage period of the bedding material since it affects the litter’s physical and handling properties such as compressibility, compaction, and cohesion [23]. Hemp shives (7.78%) showed similar moisture content to other bedding materials such as wood shavings (7.1 and 7.37%), corn stover (8.06%), rice hulls (8.37, 8.7, 10%), and wheat straw (8.44%) [23,24,25,26], but higher moisture content than recycled paper (3.82%), rice husks (4.62%), and sawdust (4.83%) [22]. 

Water absorption is an important property of bedding material as it shows the quantity of water that the material is capable of absorbing and storing. Similar results for water absorption for the ten hemp varieties were found in this study. Literature assures that hemp hurds can absorb up to five times their weight in moisture which is typically 50% higher than wood shavings [27]. Even though it was difficult to make direct comparisons with previous reports due to the different methods used and the nature of the sample, the water absorption of hemp shives in this study was lower than the only value (325.0%) found in the literature [28]. In addition, another study reported the water absorption in hemp hurds to be 356.2%. Unfortunately, we do not have any information about the nature of either of these samples to offer any further discussion. In previous studies, water absorption of bedding materials was reported and showed values of 266% for fine wood shavings, 305% for cereal straw, 320.8% for wheat straw, 330% for straw, 382% for recycled paper, 392.3% for paper cuttings, 315.9% and 460% for wood shavings, 462% for rice husks and 483% for sawdust [1,19,22,28]. This suggests that hemp shives may have a similar water absorption capacity to fine wood shavings which are known for their good moisture absorption. High water absorption is a desirable physical characteristic because it leads to the absorption of water in excreta and urine. Water absorption of hemp could even be improved if the processing method changes, for example, by decreasing the particle size. Even though the capacity to absorb water is an important value, it differs from the capacity to absorb urine, which could be higher or lower depending on the bedding material [1]. 

As happened with water absorption, ammonia absorption was also a difficult parameter to compare within the literature because of differences in methods. A high concentration of ammonia inside the animal house could represent a potential health hazard to humans and animals [29], so it is better that the bedding material has a good capacity of ammonia absorption to avoid compromising the animal health status. A significant difference was found between varieties, with Ferimon showing the highest ammonia absorption. Our results were close to those of Airaksinen et al., who reported that the relative ammonia absorption of hemp was 60%, and stated that hemp had a better ammonia absorption capacity than wood shavings (44%), and straw (4%) [8]. However, that study used horse urine in its procedure and the nature of the hemp sample was not described. Ammonia absorption is important both in summer when the indoor temperature in the animal house rises, and in winter, when the ventilation has to be reduced because of the cold. Knowing that ammonia emissions coming from animal manure is a great source of atmospheric ammonia [30], this parameter could be improved by raising the water content, shredding, or the addition of an ammonia absorbent such as sodium bisulfate [1].

Considering that this study shows a general overview of hemp hurds in the form of shives as bedding material, it is clear that some other parameters still need to be determined to provide a complete assessment of this new and still little-known product. More physical parameters such as particle size, bulk density, and water-holding capacity need to be covered. As animal health and welfare are important to consider too, microbial quality, dustiness, and more ammonia tests should be performed. The availability and price of hemp hurds also need to be studied. For this reason, the amount of bedding use per animal per day, and an economic study would be useful. In Italy, commercial brands of pure hemp shavings are sold at a range of prices from 1.4 to 3.8 euros per kilogram, depending on the quantity of the material and the target animal. Those for pets like rabbits and other rodents are always more expensive than those for horses or larger animals. Finally, to complete the zero-waste cycle of hemp, the possibility of making compost or other bio-fuel products after the farm cycle finishes needs further investigation. 

## 5. Conclusions

In conclusion, the physical characteristics of hemp shives give them the ability to become a good animal bedding material. Any of the ten hemp varieties studied could be used, alone or as a mixture, as there was no wide variation among them apart from the ammonia absorption. In addition, a comparison with other studies indicates that hemp has similar water and ammonia absorption capacities to other commercial bedding materials like wood shavings. In the future, the impact of hemp will need to be evaluated through on-farm trials.

## Figures and Tables

**Figure 1 animals-13-00284-f001:**
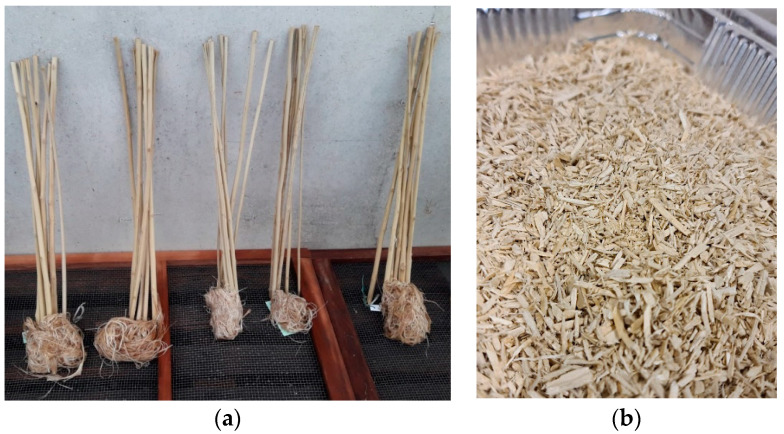
Hemp sample treatments, (**a**) fiber removal; (**b**) hemp shives.

**Table 1 animals-13-00284-t001:** LS means of physical properties of hemp shives.

Variety	Fibranova	Codimono	USO31	CS	Futura 75	E. Campana	Carmaleonte	Felina 32	Santhica	Ferimon	*p*-Value
M	7.97	7.70	8.21	7.70	7.73	7.72	7.64	7.52	7.69	7.91	0.5946
WA	250.4	212.3	281.5	235.1	235.4	211.4	310.0	317.9	234.2	231.1	0.2001
AA	45.0	53.2	51.7	51.0	49.1	47.6	49.7	48.9	48.6	55.5	0.0452

M: moisture content; WA: water absorption; AA: ammonia absorption.

## Data Availability

No new data were created or analyzed in this study. Data sharing is not applicable to this article.

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
