# Peer review of "Physical Characterization of Ten Hemp Varieties to Use as Animal Bedding Material"

_animals, 2023, doi:10.3390/ani13020284_

Round 1

Reviewer 1 Report

How it can be concluded that Hemp has the same abilities 

to that of the wood shavings or such materials when

 we don't have conducted such experiments.

Secondly, the most important thing is the evaluation in 

field conditions. This is a simple lab based 

chemical analysis, not based on actual field trials. 

Overall data is also of very little magnitude...

Reviewer 2 Report

Overall, this paper is concise and discusses the characteristics of hemp as bedding for livestock. I do have a few comments for changes to be made to the paper.

L27: Change "such as" to "including", as you analyzed each of there characteristics and no others.

L33: Change "." to "," between varieties and resp.

L39: Remove the space between "live" and "stock".

L42: Remove "some".

L100: Please provide a brief description of the WA and AA analyses. As you indicate these are different from other studies in your discussion, having an idea of what you did would be helpful without looking up the papers.

L123: From "the whole" to the end of the sentence is confusing. Suggest rewording this.

L126: Change "use" to "used"

L140: Need to indicate that you're discussing the difference between the varieties here. Suggest changing to "not significantly different".

L166: Change "be even" to "even be"

L170: Need to use abbreviations consistently throughout the paper or just not use them. 

L173: This citation is not consistent with others in the paper.

L175: For M and WA, you discuss comparisons even though to bring up differences in methodologies. You should also do that for AA. You say that another paper found AA similar to wood shavings and straw but do not indicate those values. Please include other relevant comparisons, even with different methodologies. 

L175: You indicate that horse urine is different than in your study; however, the methodology you cited in your methods used horse urine. What type of urine was used in your study?

L186: Change "Last, but not least" to "Also", so similar. This is more formal and you have a "finally" later in the paragraph, which means this is not last.
